# Experimental Study on Evolution of Chemical Structure Defects and Secondary Contaminative Deposition during HF-Based Etching

Xiao Shen [1,2,3], Feng Shi [1,2,3,*], Shuo Qiao [1,2,3], Xing Peng [1,2,3] and Ying Xiong [1,2,3]

1 College of Intelligence Science and Technology, National University of Defense Technology, Changsha 410073, China; hillbert2009@163.com (X.S.); sqiao525@163.com (S.Q.); pengxing@fudan.edu.cn (X.P.); annie_xiong@163.com (Y.X.)
2 Hunan Key Laboratory of Ultra-Precision Machining Technology, National University of Defense Technology, Changsha 410073, China
3 Laboratory of Science and Technology on Integrated Logistics Support, National University of Defense Technology, Changsha 410073, China
* Correspondence: shifeng@nudt.edu.cn

**Abstract:** Post-processing based on HF etching has become a highly preferred technique in the fabrication of fused silica optical elements in various high-power laser systems. Previous studies have thoroughly examined and confirmed the elimination of fragments and contamination. However, limited attention has been paid to nano-sized chemical structural defects and secondary precursors that arise during the etching process. Therefore, in this paper, a set of fused silica samples are prepared and undergo the etching process under different parameters. Subsequently, an atomic force microscope, scanning electron microscope and fluorescence spectrometer are applied to analyze sample surfaces, and then an LIDT test based on the R-on-1 method is applied. The findings revealed that appropriate etching configurations will lead to certain LIDT improvement (from initial 7.22 J/cm$^2$ to 10.76 J/cm$^2$), and HF-based etching effectively suppresses chemical structural defects, while additional processes are recommended for the elimination of micron- to nano-sized secondary deposition contamination.

**Keywords:** HF-based etching; micro morphology; chemical structure defects; depositive reaction products

## 1. Introduction

As the crucial terminal part of the ITER (International Thermonuclear Experimental Reactor), Final Optics Assembly (FOA) comprises fused silica optical elements capable of withstanding immense laser energy fluxes approaching the mega-joule level [1]. In operation, multiple laser beams traverse the FOA and converge upon a polyimide target capsule to necessitate the prior thermonuclear reaction, which induces subsequent nuclear fusion. In the National Ignition Facility (NIF) of the United States, 192 laser beams simultaneously transmit through the FOA, achieving an output energy of up to 1.9 mega-joules [2]; in France's Laser Megajoule facility (LMJ), the number of laser beams increases to 240, reaching a total energy output of 2.4 mega-joules [3]. Exposed to such high throughput, fused silica optical elements are vulnerable to various laser induced damages that can potentially result in rapid component failure.

A previous study revealed that the intrinsic LIDT (Laser-Induced Damage Threshold) of fused silica stands at 100 J/cm$^2$ [4]; however, in practice, the observed LIDT of fused silica optical components is notably lower than theoretical threshold. Numerous researchers have identified the primary factors contributing to this significant discrepancy between practical and theoretical LIDT values, such as damage precursors [5,6] derived from manufacturing

integrated with grinding, polishing and post-treatment processing, including fragments, contamination and nano-scale defects. These defects exhibit a strong propensity to induce intense laser radiation absorption, resulting in significant energy accumulation, and once the energy deposition surpasses a certain limit, severe laser-induced damage ensues. Of all the precursors, fragments derived from material cutting, grinding and polishing are often exhibited as surface or subsurface cracks, pits, scratches and pocking marks, etc. Typically, the sizes of fragments range in size from sub-micron to several nanometers [7,8]; the occurrence of contamination is the result of residual polishing powder remaining in the superficial hydrolysis layer and subsurface defect layer, including particles such as $CeO_2$ and $ZrO_2$ during polishing and Cu and Fe particles during machining [9–11]. Nano-scale defects including chemical structure defects and trace amounts of salt deposits, unlike fragments and contamination, tend to cause laser-induced damage under low throughput and will evolve into precursors when exposed to high throughput. The former results from molecular bond breaking during machining [12,13], and the latter comes from post-processing aiming at removing surface and sub-surface precursors. Ultimately, all of these precursors can lead to the performance degradation of the FOA [14].

To mitigate various damage precursors, the wet etching process is also introduced [15–17]. This approach aims to suppress precursors while maintaining optical surface precision. The removal mechanism of surface and subsurface precursors during wet etching is the global chemical reaction between the etchant and the substrate. Scholars have conducted thorough research on HF-based wet etching processes; Wong investigated morphological evolution during the HF etching of cracks originating from the grinding process and found that the passivation of cracks can lead to a significant improvement in LIDT [16]; the Lawrence Livermore National Laboratory integrated HF etching with ultrasonic washing technology to establish AMP (Advanced Mitigation Processing), a method which has proven to be a reliable and effective means of removing fragments and contamination in post-treatment processes [15,18]. Through the latest AMP technology, Bude discovered that the initially scratched optical surface remained intact even under laser irradiation of 10 J/cm$^2$ (@351 nm, 5 ns) after passivation [19]. Zheng discovered that an appropriate etching process parameter can enhance LIDT and further identified a near-linear correlation between surface hardness and LIDT [20]. As research progressed, scholars also revealed that HF wet etching has a tendency to induce secondary precursors due to depositive reaction products, compromising surface accuracy if the etching parameter is not carefully controlled. Suratwala et al. observed that secondary pollutants adhere to the surface during HF wet etching, affecting the intrinsic characteristics of fused silica, including LIDT [15]. Bude revealed that nano-sized salt deposits resulting from post-treatment processes play an important role in laser-induced damage; when exposed to laser irradiation at 488 nm and 25 J/cm$^2$ for 5 ns, NaCl crystal deposits from deionized water caused significant damage to optical elements [13].

Since current post-processing based on HF etching on optical elements in high-power laser systems is a kind of "pollution first and treatment later" or "treatment during pollution" type of process in essence, the intrinsic surface of optical elements is theoretically inaccessible, and the final performance of the optical element is determined by post-processing to a large extent. However, the majority of existing studies on post-processing are primarily focused on crack passivation and pollution removal through HF etching. The impact of HF etching on chemical structure defects and the evolution and impact of secondary pollution introduced by HF etching have not been fully elucidated. Additionally, with HF being a highly toxic and corrosive compound, only by understanding the laws of the derivation and evolution of various precursors in HF etching can we achieve optical surfaces with superior performance in a more environmentally friendly and efficient manner. The paper is structured as follows: in Section 2, the sample preparation, testing, and characterization methods are introduced. Section 3 presents the research findings. Section 4 discusses the pertinent research results, and Section 5 summarizes the entire work presented in the paper.

## 2. Materials and Methods

As a weak acid, the ionization products of HF solution are $H^+$, $F^-$, $HF_2^-$ and $H_2F_2$ [21]. In HF wet etching, the corresponding chemical equation is given by the following [15]:

$$SiO_{2(solid)} + 3HF_{2(aq)}^- + H_{(aq)}^+ = SiF_{6(aq)}^{2-} + 2H_2O_{(aq)} \tag{1}$$

The mechanism of the dissolution of $SiO_2$ in HF wet etching is clarified by Judge [22]: the reaction rate is determined by the adsorption process of HF molecules, $HF_2^-$ and $H^+$ ions on fused silica. Similar to $SiO_2$ crystals, silicon atoms and oxygen atoms together form a three-dimensional reticular structure, composed of $SiO_4$ tetrahedrons through covalent bonds. During etching, $HF_2^-$ and HF are adsorbed onto siloxane groups ($\equiv$Si-O-Si$\equiv$), while $H^+$ ions are attached to bridging oxygen atoms in siloxane bonds. The gathering effect of HF and $HF_2^-$ leads to an increase in electron density on bridging oxygen atoms, which strengthening the alkalinity of these oxygen atoms; subsequently, more $H^+$ ions are attracted, resulting in an elevated rate of silicon–oxygen bond breaking. The entire process encompasses two distinct steps, as outlined below:

Step one, the protonation of bridging oxygen atoms:

$$Si - O - X + H^+ \rightarrow Si - O(H)^+ - X \text{ (X represents Si or H)} \tag{2}$$

Step two, the nucleophilic attack of electrophilic silicon atoms by $HF_2^-$:

$$Si - O(H)^+ - X + HF_2^- \rightarrow Si - F + HO - X + HF \tag{3}$$

The etching rate model can be defined as follows [21]:

$$R = \left(k_1\left[HF_2^-\right] + k_0[H_2F_2]\right)\frac{K_3\left[H^+\right]}{1 + K_3\left[H^+\right] + 1/K_4\left[H^+\right]} + k_2\left[HF_2^-\right]\frac{K_4\left[H^+\right]}{1 + K_4\left[H^+\right] + K_3K_4\left[H^+\right]^2} \tag{4}$$

The dissolution rate of $SiO_2$ in hydrogen fluoride solution is governed by the protonation and deprotonation of the surface reaction site; for a complete $SiO_4$ element, four rapid nucleophilic substitution reactions are essential to deprive one silicon atom away from the substrate. Therefore, it can be concluded that the presence of numerous chemical structure defects on fused silica substrate will certainly lead to an accelerated etching rate.

### 2.1. Sample Preparation

To study the evolution of damage precursors in HF etching, 6 samples are prepared, marked from #0 to #5, and the size of each sample is Ø 50 mm × 5 mm. The raw material of each sample is Suprasil 300 from the Heraeus company. Before etching, each sample goes through a smoothing process on a self-developed system [23] via pitch lap; the detailed parameters are shown in Table 1. Measurement via a Zygo interferometer indicates that the initial surface roughness is 0.28 nm, as illustrated in Figure 1.

**Table 1.** Smoothing parameters.

| Parameters | Value |
|---|---|
| Polishing abrasive | $CeO_2$ |
| Abrasive diameter | 1.5 μm |
| Rotating rate | 150 r/min |
| Feeding rate | 300 mm/min |
| Smoothing pressure | 0.02 Mpa |
| Smoothing duration | 180 min |

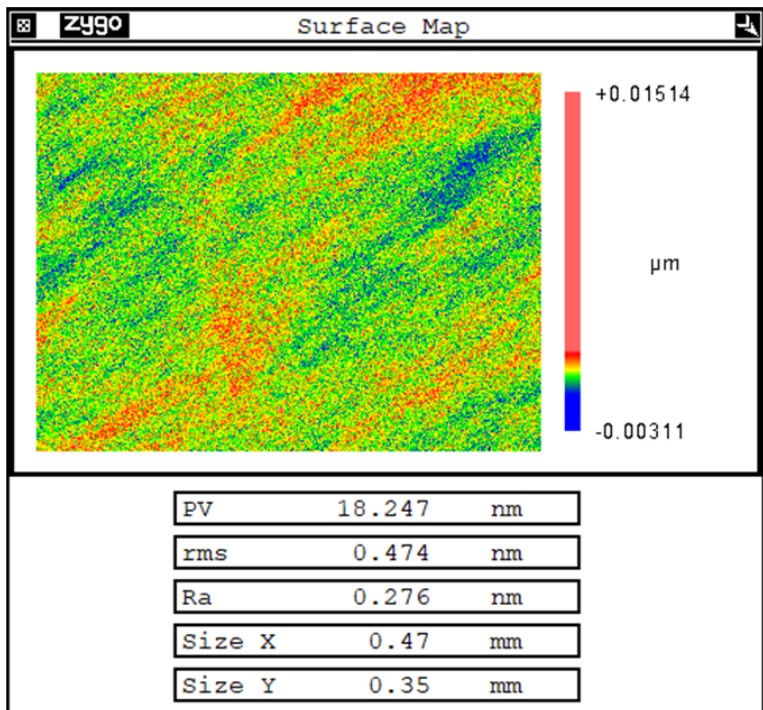

**Figure 1.** Surface measurement of initial samples.

*2.2. Megasonic-Aided Etching Configuration*

Current HF etching applications are usually performed in two ways: dynamic etching (or megasonic-aided etching) and static etching. In the extensive removal process via static etching technology, the rapid accumulation of reaction products leads to significant local precipitation easily, causing a profound effect on removal homogeneity and resulting in the deterioration of surface shape and roughness; consequently, the LIDT decreases drastically [24,25]. To avoid this problem, megasonic-aided etching, which combines the etching process with ultrasonic vibration, are more appealing to researchers. Assisted by an ultrasonic acoustic field, a thin boundary layer is generated adjacent to the reaction surface, as shown in Figure 2. According to the theory of boundary layer, the liquid vibration and surface material transfer efficiency caused by liquid vibration exhibit a negative correlation with the thickness of the acoustic layer. The thickness of the acoustic layer is determined by Equation (5), as stated in reference [21].

$$\delta = \sqrt{\frac{\mu}{\pi f \rho}} \tag{5}$$

where $\mu$ and $\rho$ denote the viscosity and density of the liquid, respectively, and $f$ represents the frequency of the megasonic.

In the process of megasonic-aided etching, the mass fraction of the HF solution is 5 wt. %, and the frequency is set to 1.3 MHz to ensure that the thickness of the acoustic layer caused by fluid viscosity remains below 0.5 µm [22]. Influenced by an ultrasonic acoustic field, the diffusion of micro reaction product deposition in solution will be more efficient, leading to a significant enhancement in the reaction rate between HF and $SiO_2$. For the 6 samples considered in this paper, Figure 3 illustrates the dynamic etching procedures and etching configurations. Prior to HF etching, deionized water rinsing is able to eliminate any disruptive contamination introduced by previous machining steps. The specific ultrasonic parameter setup for the rinsing is detailed in Table 2. All operations outlined in Figure 3 are carried out in a class 100 clean room, with a steady ambient temperature of 25 °C, to avoid external contamination during rinsing, etching and drying.

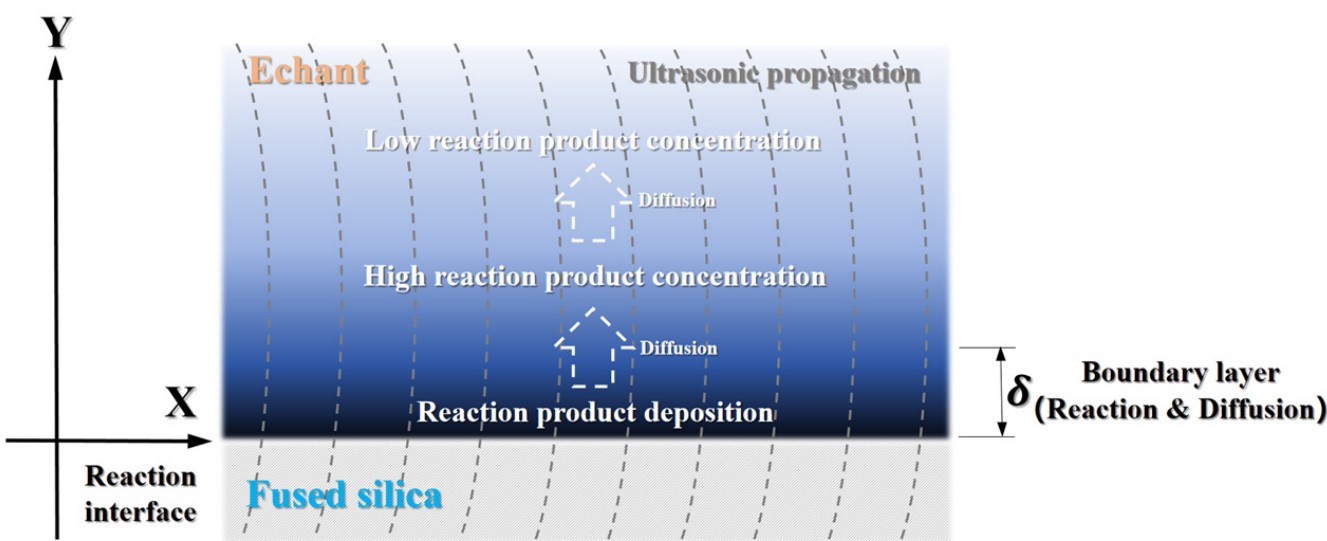

**Figure 2.** HF etching aided by megasonic acoustic field.

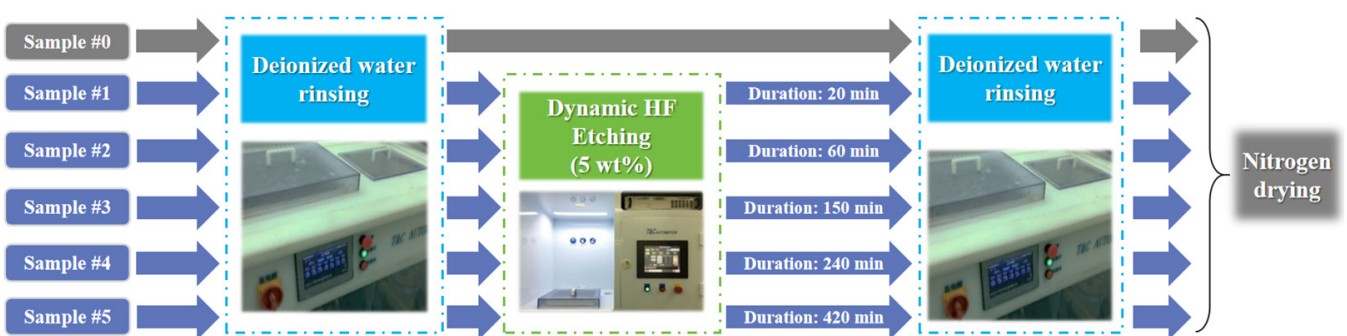

**Figure 3.** Etching schedule of each sample.

**Table 2.** Rinsing parameters.

| Step | 1 | 2 | 3 | 4 | 5 | 6 | 7 |
|---|---|---|---|---|---|---|---|
| Frequency [kHz] | 50 | 100 | 150 | 200 | 250 | 300 | 350 |
| Duration [min] | 5 | 5 | 5 | 5 | 5 | 5 | 5 |
| Temperature [°C] | | | | 22 | | | |

*2.3. Laser-Induced Damage Threshold Test Configuration*

The laser induced damage test for each sample is carried out in the laboratory of advanced optical fabrication, National University of Defense Technology, and the test platform and the schematic are shown in Figure 4a,b, respectively. The test wavelength is 355 nm, the pulse width (FWHM) is 7 ns, the target spot shape is round, the laser spot diameter is 1.2 mm, and the degree of modulation is 3.2. During the test, the temperature is $22 \pm 0.2$ °C, and the humidity is $35 \pm 5\%$. The measurement of threshold is based on the R-on-1 method; namely, by testing the same point with increasing laser flux, once laser damage occurs, the corresponding throughput is the local LIDT of the point.

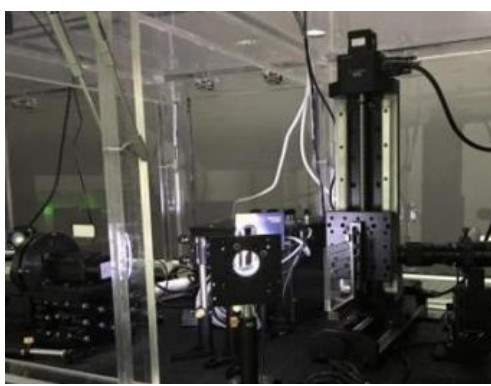

(**a**) Damage test platform

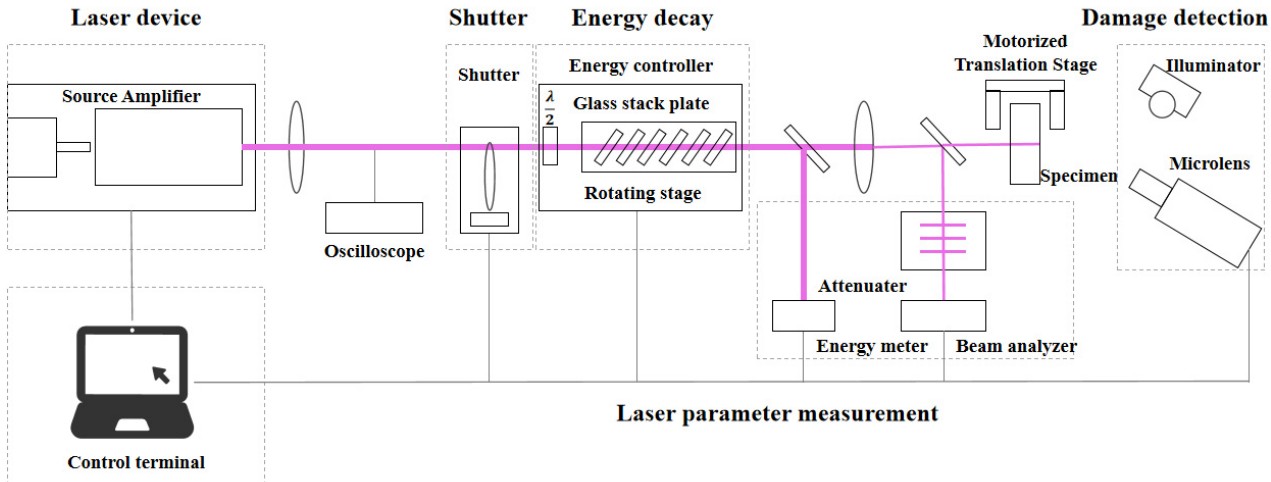

(**b**) Schematic of the platform

**Figure 4.** LIDT test platform.

## 3. Results

### 3.1. Evolution of Surface Micro Morphology

The atomic force microscope (AFM) of Bruker (detailed configurations are scanning frequency of 1.0 Hz and measuring area of 5 μm × 5 μm) is applied to assess the surface quality. Results on sample #0 revealed evident polishing marks left by the prior smoothing process on the initial surface, as depicted in Figure 5a. Additionally, a Form Talysurf PGI 1240 surface profiler is employed to detect the real-time etching depth.

In contrast, sample #0 remains unexposed to dynamic etching, maintaining its original roughness. Figure 5b–f depicts the microstructural evolution under varying etching durations, in 20 min etching period increments. Following the removal of the superficial hydrolytic layer, the numbers of subsurface defects are revealed. These defects, including scattered micro black pits and scratches, become evident, as shown in Figure 5b. As etching proceeds, the micro pits become increasingly shallower, and mottled gel-like depositions begin to emerge, appearing as white spots in the AFM images shown in Figure 5b–f. The figure also reveals that the quantity of white deposition spots peaks at 150 min of etching (sample #3), and subsequently, the count of white deposition spots decreases. Concurrently, the initial roughness deteriorates, increasing from 0.28 nm to 1.5 nm.

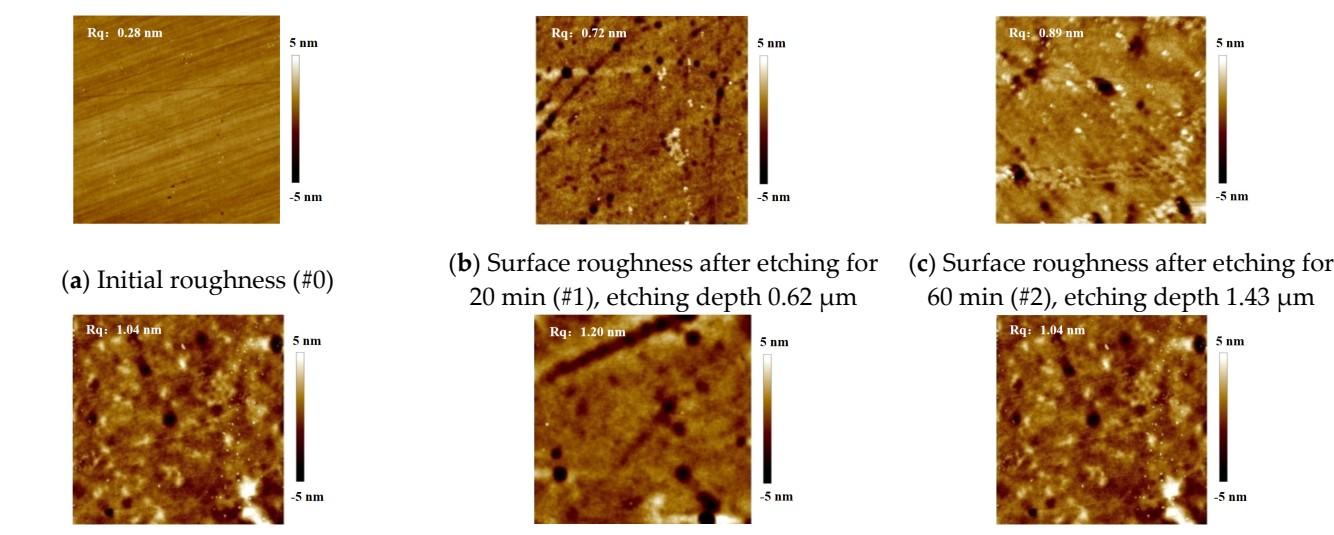

(**a**) Initial roughness (#0)

(**b**) Surface roughness after etching for 20 min (#1), etching depth 0.62 μm

(**c**) Surface roughness after etching for 60 min (#2), etching depth 1.43 μm

(**d**) Surface roughness after etching for 150 min (#3), etching depth 3.59 μm

(**e**) Surface roughness after etching for 240 min (#4), etching depth 5.47 μm

(**f**) Surface roughness after etching for 420 min (#5), etching depth 9.43 μm

**Figure 5.** Micro morphology of the samples under different etching depths.

### 3.2. Composition Identification of Micro Depositions

Using a PHENOM-ProX scanning electron microscope, a detailed examination is conducted on mottled sub-micron-scale residual depositions on sample #5. The elemental composition of two separate test points and surrounding areas are analyzed with an element mapping module. Figure 6b–e presents the energy spectrum scanning data of test point one, and it is evident that the distribution of impurity elements Na, Ca, Cl and C align precisely with the morphological distribution of residual depositions at test point one; the detailed composition ratio for each element is shown in Table 3.

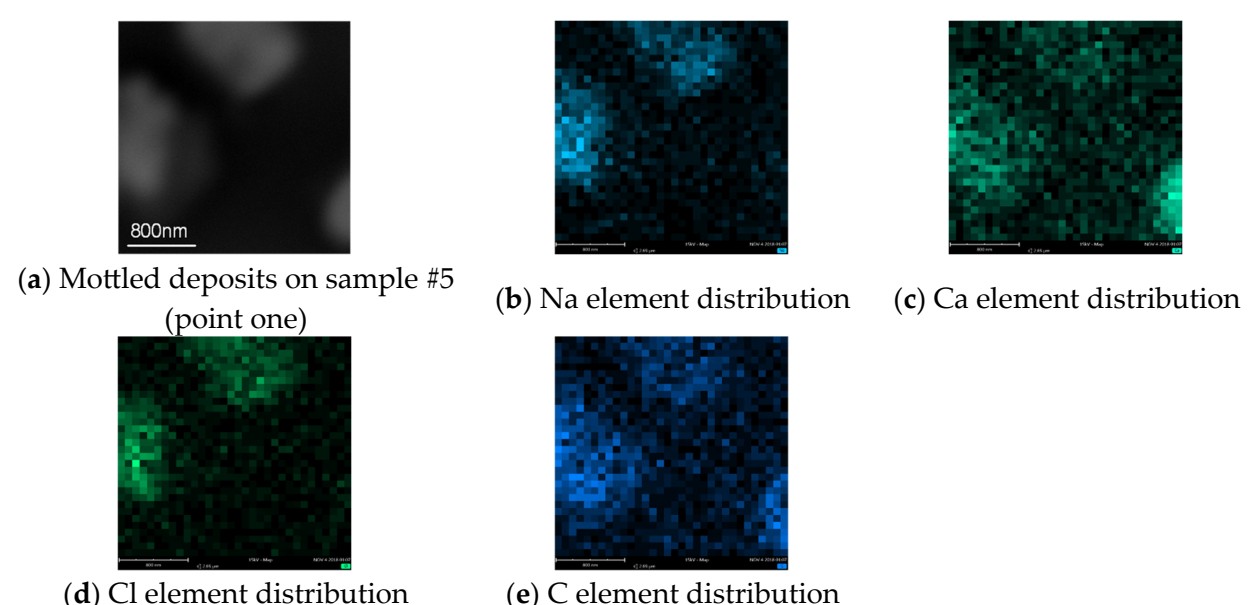

(**a**) Mottled deposits on sample #5 (point one)

(**b**) Na element distribution

(**c**) Ca element distribution

(**d**) Cl element distribution

(**e**) C element distribution

**Figure 6.** Energy spectrum scanning results of test point one.

**Table 3.** Elemental composition of the deposits at test point one.

| Element | Si | O | Na | C | Cl | Ca | Others |
|---|---|---|---|---|---|---|---|
| Composition ratio | 50.48% | 18.42% | 1.13% | 27.87% | 0.51% | 0.11% | 1.48% |

Under same image scale parameters, test point two is also observed, and the corresponding energy spectrum scanning results in Figure 7 demonstrate a shared distribution among residues and Na, K, Cl and C elements.

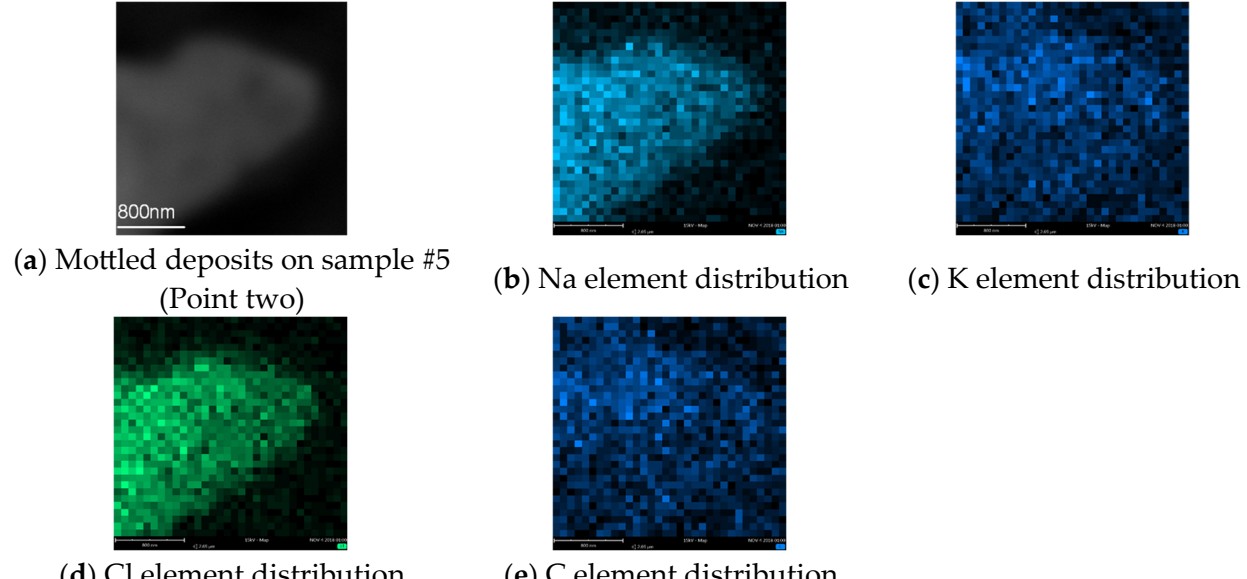

(**a**) Mottled deposits on sample #5 (Point two)

(**b**) Na element distribution

(**c**) K element distribution

(**d**) Cl element distribution

(**e**) C element distribution

**Figure 7.** Energy-spectrum scanning results of test point two.

Considering results above, besides Si and O elements originating from sample substrate, the residual deposition also contains C, Na, Cl, K and other elements, as shown in Tables 3 and 4. For both points, the C element content is significantly higher than other external elements. Furthermore, it is worth mentioning that the gathering-up effects of impure elements on residual depositions are particularly evident at both test points when compared to surroundings.

**Table 4.** Elemental composition of the deposits at test point 2.

| Element | Si | O | Na | C | Cl | K | Others |
|---|---|---|---|---|---|---|---|
| Composition ratio | 15.62% | 39.33% | 3.76% | 35.48% | 2.10% | 0.24% | 3.47% |

### 3.3. Evolution of Chemical Structural Defects

In nano-scale fused silica substrate, there appears to be a 3D reticular amorphous structure composed of countless silicon–oxygen bonds. Previous research revealed that the chemical structure defects resulting from pre-machining processing are an oxygen-deficient center (ODC), a nonbridging oxygen hole center (NBOHC) and an E'-center (E'), as depicted in Figure 8a–c respectively. These imperfections, stemming from disrupted silicon–oxygen bonds, exert a significant constraint on the LIDT of optical elements [26,27].

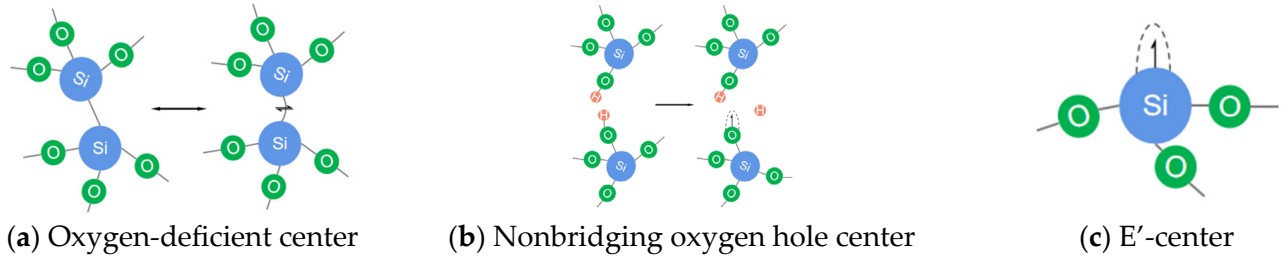

(**a**) Oxygen-deficient center

(**b**) Nonbridging oxygen hole center

(**c**) E'-center

**Figure 8.** Chemical structure defects in fused silica substrate.

Chemical structural defects can be characterized via fluorescence spectroscopic detection. In this regard, an AJY TAU-3 fluorescence spectrometer is applied to test samples #0, #3 and #5. The testing parameters are specified as follows: the excitation wavelength is 248 nm, the spectral resolution is 0.01 nm, and a long-wavelength pass filter with a cutoff wavelength of 320 nm is also used to mitigate the interference of secondary excitation.

The fluorescence spectra of samples #0, #3 and #5 are presented in Figure 9. The initial surface exhibits two prominent absorption peaks. The peak centered around 440 nm originates from ODC defects, whereas the peak centered around 640 nm corresponds to NBOHC defects [28]. These structural defects arise from prior machining processes. During hydrolysis, the nano network of fused silica is broken first, enabling impurity elements such as Ce and H to penetrate the surface of the fused silica and form new bonding structures with Si and O elements; hence, ODC and NBOHC defects are introduced into sample #0. As depicted in Figure 9b, after 150 min, the fluorescence spectrum of ODC on sample #3 exhibits a significant reduction in characteristic peaks, and the intensity decreases from 1514.3 to 1031.0, which indicates a considerable decrease in the concentration of ODC defects; for NBOHC, the intensity level around 640 nm remains. However, the intensity level of noise near two characteristic absorption peaks changes significantly after etching; the reason is as follows: with the deprivation of oxygen atoms and silicon atoms from three-dimensional reticular structure during a chemical reaction, the residual part of an formerly complete $SiO_4$ tetrahedron will continuously induce new chemical structure defects; thus, the noise intensity level is strengthened.

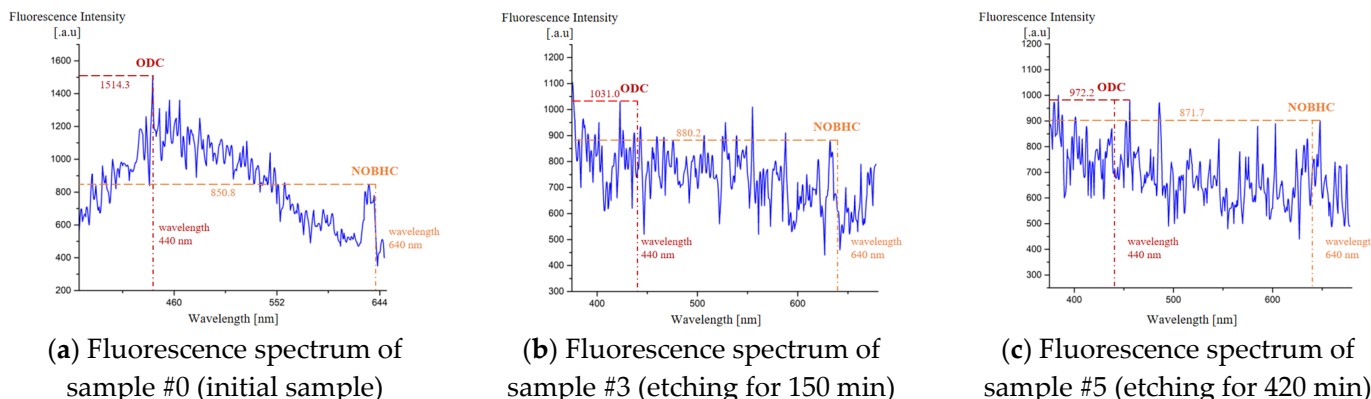

(**a**) Fluorescence spectrum of sample #0 (initial sample)

(**b**) Fluorescence spectrum of sample #3 (etching for 150 min)

(**c**) Fluorescence spectrum of sample #5 (etching for 420 min)

**Figure 9.** Fluorescence spectrum of the samples under different etching durations.

Upon further etching to 420 min, the fluorescence spectrum for #5's surface remains consistent with that of sample #3, which means that even excessive HF etching does not introduce any new characteristic peaks sourced from ODC. For NBOHC, the situation is quite different, as the intensity level indicates no noticeable change. Based on the results, HF etching technology exhibits a significant suppressive effect on ODC defects for fused silica-based optical elements.

*3.4. LIDT Testing Results*

The results of laser damage threshold testing on different HF etching depths are presented in Figure 10. After the polishing process, the initial LIDT is 7.22 J/cm$^2$. As the etching depth increases to 3.53 μm, the LIDT increases to 10.76 J/cm$^2$, which exhibits a notable upward trend. However, further increases in etching depth result in a significant decrease in LIDT. Specifically, at etching depths of 5.46 microns and 9.54 microns, the LIDT decreases by 13.75% and 19.61%, respectively, compared to the highest value ever achieved.

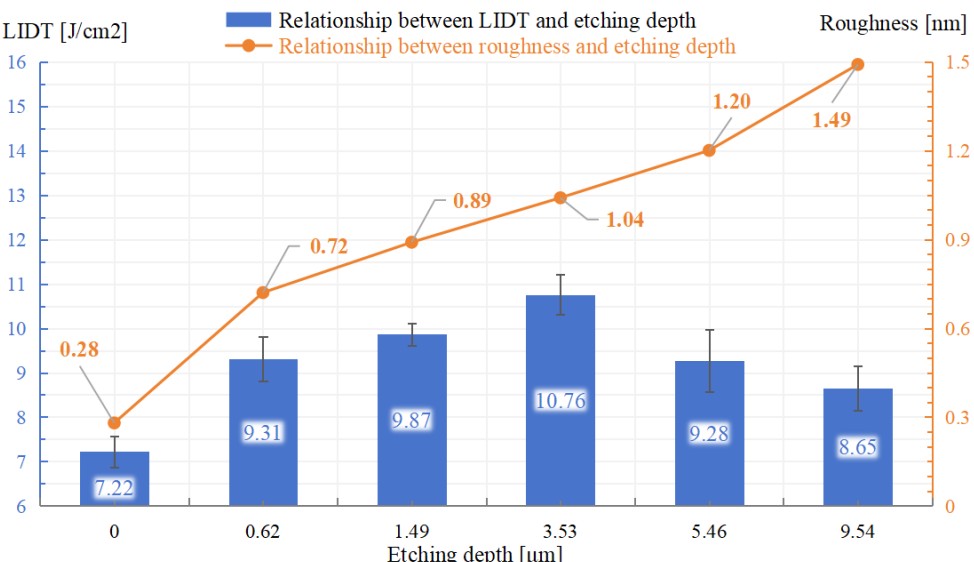

**Figure 10.** Relationship between LIDT, surface roughness and etching depth.

## 4. Discussion

Through measurements and LIDT tests in Section 4, the specific influence of hydrogen fluoride-based etching on the characteristics of a fused silica sample are revealed. Nonetheless, the underlying mechanism of phenomena observed during the tests merit further elucidation, and the enhancement of laser damage resistance performance also merits deeper exploration.

Over a duration of 420 min, the average removal rate is approximately 23 nm/min, compared to that of 30 to 170 nm/min in the grinding process and 2 to 8 nm/min in the polishing process [29]. Hydrogen fluoride-based etching is of high efficiency due to its feature of a global reaction effect with any exposed surface of the substrate.

The relationship between roughness and etching depth is also shown in Figure 11. Surface roughness decays rapidly first and then slows down, keeping a relatively stable decay rate, followed by a gradual slowing down, then maintains a relatively constant decay rate. Meanwhile, the etching rate exhibits a similar trend. This phenomenon can be attributed to the specific composition of the superficial hydrolysis layer (presented in Figure 12) on initial samples; the layer consists of an inhomogeneous and loosely packed mixture of micro fused silica fragments, silicic acid gel (hydrolysis relevant product, $\equiv$Si—OH) and impurities originating from the deposition of hydrolysis during prior polishing and smoothing processes. The normal compression loads on a fused silica surface during these prior machining processes will enhance the chemical activity of the hydrolysis layer [30]. Additionally, some impurities do not react with HF. The combined effect of all these factors will certainly result in a rapid reaction rate between the hydrolysis layer and HF solution. Once the hydrolysis layer is removed, the reaction rate slows down.

Meanwhile, the etching product also deposits on an uneven reaction surface, presented as micron-scale white spots in Figure 5. Then, HF has to penetrate these gel-like depositions to react with the substrate, which leads to the further deterioration of surface roughness. It is also noteworthy that the etching rate within the hydrolysis layer exhibits a non-constant characteristic. The gradually decreasing roughness decay rate and etching rate observed in the figure suggest that the concentration of the hydrolysis layer diminishes with the increase in layer depth. As the etching process progresses, subsurface defects resulting from the prior processing stage are exposed, the surface becomes rougher and scratches transform from sharp contours to smoother arc-shaped contours. The increasingly rough micro morphology of the local surface gives rise to varying local etching rates. Additionally, the accumulation of a depositive etching product manifests as micron-scale white spots in Figure 5. On an uneven reaction surface, the homogeneity of the local reaction rate will be

even worse, since HF has to penetrate these gel-like depositions to react with the substrate. This phenomenon consequently leads to a further deterioration in surface roughness.

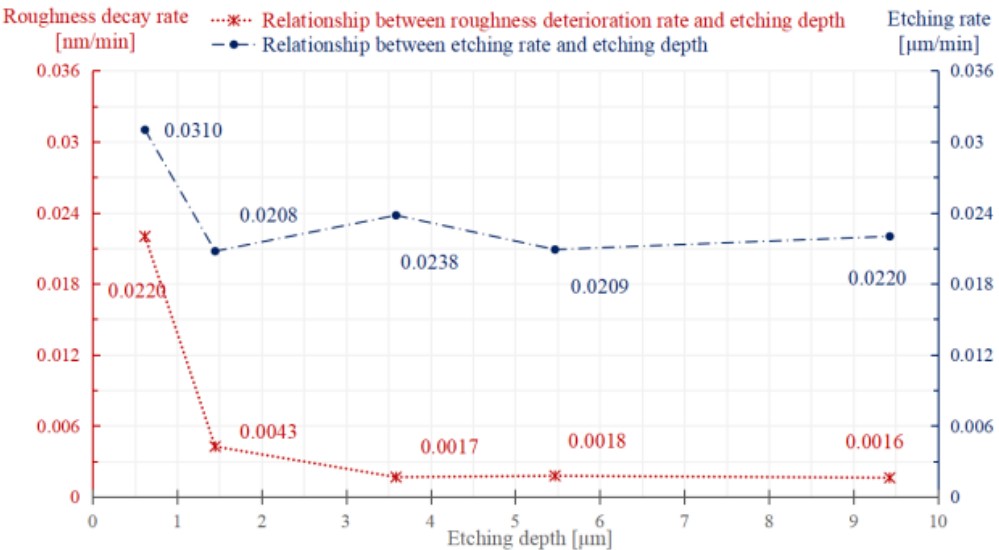

**Figure 11.** Relationship between roughness and etching depth.

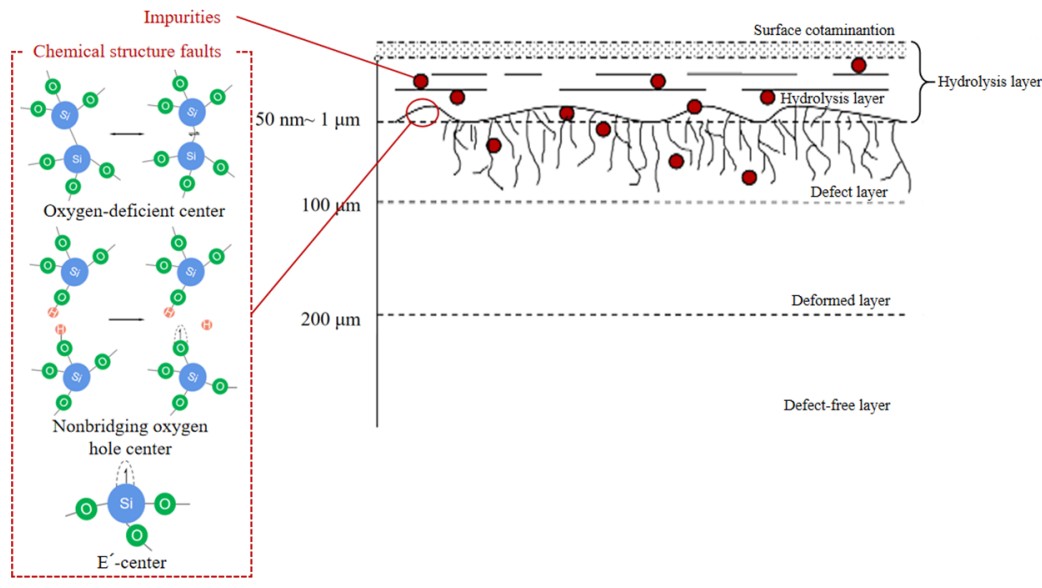

**Figure 12.** Specific composition of superficial hydrolysis layer.

Despite the application of megasonic-assisted etching, white mottled residual depositions can still be observed on sample #5 through SEM. These depositions are primarily composed of sodium, calcium and potassium salts, and of all the elemental constituents, silicon and oxygen elements originating from fused silica substrate dominate the elemental composition. Notably, a significant accumulation of impure elements exists within the residual depositions adhering to the surfaces, which is attributed to the adsorption of ions through a gel-like depositive reaction product during etching. The impure elements within these depositions may originate from two potential sources:

(1) From fused silica samples: in HF etching, the chemical network structure of silicane will be disassembled, causing the dissolution of the alkalis initially embedded in fused silica. For example, the contents of calcium, potassium and sodium elements in suprasil 300 produced by Heraeus that are used in this paper are 0.05 ppm, 0.01 ppm and 0.05 ppm, respectively [31];

(2)　From deionized water: the alkali elements may also stem from deionized water in HF solution, as deionized water often contains trace amount of dissolved ionic impurities.

By analyzing the LIDT test results, it is evident that the appropriate selection of etching parameters can significantly enhance the LIDT. During first 150 min of etching (where etching depth reaches 3.59 μm), HF effectively reacts with the fused silica, removing fragment defects, chemical structure defects and impurities resulting from prior polishing and smoothing processes, leading to a certain improvement on LIDT. However, once the etching depth exceeds 3.59 μm (over 150 min), even with small alterations in surface roughness, a notable reduction in LIDT is observed. A phenomenon manifests where the negative impacts of reaction product deposits start to affect LIDT; additionally, excessive HF etching will also cause the severe deterioration of surface roughness. Both factors hinder the practical performance improvement of optical elements in high-power laser systems. In that sense, single HF etching alone is insufficient as a complete precursor treatment for fused silica optical elements to obtain the desired defect-less surface. For further LIDT enhancement, the implementation of additional processes such as IBF [32] and KOH etching [33] would be highly beneficial, particularly in mitigating residual deposits.

## 5. Conclusions

To explore the evolution of chemical structural defects and micron- to nano-sized secondary contaminative deposition during HF etching, the authors conducted a set of experimental research, and the relevant conclusions are as follows:

(1)　During etching, the surface roughness deteriorates steadily; as the etching continues, white mottled gel-like depositions emerge. The roughness deterioration rate illustrates that the concentration of the hydrolysis layer diminishes with the increase in layer depth, and the depositions also add to further surface roughness deterioration;

(2)　For chemical structure defects elimination, HF exhibits a significant suppressive effect on ODC defects for fused silica-based optical elements;

(3)　The white mottled depositions consist of Na, K, Ca, Cl and C elements originating from either substrate or deionized water. The precisely shared shape between deposition and impure elements distributions illustrate that during etching, the impure ions are absorbed and deposited with the reaction product. For the thorough removal of impure elements, additional processes are suggested.

**Author Contributions:** Conceptualization, X.S.; methodology, X.S., S.Q. and X.P.; software, X.S.; validation, X.S., X.P. and Y.X.; formal analysis, X.S. and S.Q.; investigation, X.S.; resources, X.S.; data curation, X.S.; writing—original draft preparation, X.S.; writing—review and editing, X.S. and S.Q.; visualization, X.S.; supervision, F.S.; project administration, F.S.; funding acquisition, F.S. All authors have read and agreed to the published version of the manuscript.

**Funding:** This work is supported by the Strategic Priority Research Program of the Chinese Academy of Sciences (No. XDA25020317) and the Science and Technology Innovation Program of Hunan Province (2022RC1138, 2023JJ30079).

**Institutional Review Board Statement:** Not applicable.

**Informed Consent Statement:** Not applicable.

**Data Availability Statement:** The original contributions presented in the study are included in the article, further inquiries can be directed to the corresponding author.

**Conflicts of Interest:** The authors declare that there are no conflicts of interest.

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
