# Peer review of "Experimental Study on Evolution of Chemical Structure Defects and Secondary Contaminative Deposition during HF-Based Etching"

_photonics, doi:10.3390/photonics11050479_

Round 1

Reviewer 1 Report

Comments and Suggestions for Authors

In the paper, the authors focus on nano-sized  defects and secondary precursors during etching, relevant tests and analysis are conducted. Surface of fused silica samples under different processing parameters are analyzed by AFM, SEM, and fluorescence spectrometry, chemical structure defects are significantly suppressed, subsequent LIDT test verified that appropriate etching configuration will improve the laser resistance performance. Additionally, a particular phenomenon that reaction product deposits and element contaminants shares a common shape is also revealed. Fig 12 and Fig 11 are the same picture, please check and replace with the correct one.In Fig 11 there’s a notable etching rate decrease after first 20 minutes, the paper attributes this phenomenon to removal of the hydrolysis layer, then how thick is the hydrolysis layer in normal processing empirically?

Comments on the Quality of English Language

Polish your English.

Author Response

Thank you for your work for our manuscript. Those comments are all valuable and very helpful for revising and improving our paper. We have studied comments carefully and modified the manuscript accordingly. Detailed corrections are listed below point by point:

  1. Fig 12 and Fig 11 are the same picture, please check and replace with the correct one.

The images of Fig 12 and Fig 11 are carefully checked, the formerly wrong image in Fig 12 is replaced with the correct one, thank you for pointing out my fault!

  1. In Fig 11 there’s a notable etching rate decrease after first 20 minutes, the paper attributes this phenomenon to removal of the hydrolysis layer, then how thick is the hydrolysis layer in normal processing empirically?

Dear reviewer, for hydrolysis layer, the empirical removal duration is usually less than few hundreds of nanometers, which is consistent with conclusion with reference [1] and [2].

[1] Cheng Jian, et al. Experimental study on HF etching of fused silica optical elements[J]. High power laser and particle beams, 2017, 29(11): 7.

[2] Zhong Yaoyu. Research on damage precursor characterization and mitigation of high-performance ultra-violet laser irradiated fused silica optics[D], University of national defense, 2018.

Reviewer 2 Report

Comments and Suggestions for Authors

Interesting result of LIDT improvement - not unexpected when considering the number of publications / products with sub-wavelength structures & motheyes  and their LIDT gains. This paper tries to justify the change in chemical terms, not electric field or rigidly coupled wave analysis.

The paper has many required changes, I'll list the majority but strongly recommend going to review again before publication.

Line 40 practice in place of practical

Line 50 sub-micron and hundreds of nm is the same thing, this makes no sense

Line 53 subscript after O in CeO2 ZrO2

Line 78 LIDT not LITD

Line 100,102,105,106,108,113,115,116 subscript

Table 1 subscript

Line 135 LIDT not LITD

Figure 2 writing too small to read

Line 151 subscript

Figure 3. Writing too small in arrows

Table 2. kHz not KHZ

Line 168 LIDT not LITD

and Figure 4 title

Figure 4b font too small to read

Line 175 Bruker not Burker

Line 178 bad English suggest Additionally not Besides

Figure 6b and Figure 7b unclear, suggest a table is better

Line 219 LIDT not LITD

Figure 10 scale in y-axis doesn't show results well, suggest broaden and rescale

Line 246 refers to figure 12, suggest this should be figure 10

Line 260 remove "the whole duration" as poor English

Line 260/261 suggest change to nm/min to make more readable

Line 336 molted? Should this be mottled?

Lines 352 to 361 need to be removed, this isn't your paper

Line 363 should be Title Case for Authors not CAPITALS

Line 402 the same

Why are there [C,J,D,M] in brackets? Not a standard format

Comments on the Quality of English Language

The English does not always read well and recommend another review round after re-write.

Author Response

Dear reviewer, thank you for your work for our manuscript. Those comments are all valuable and very helpful for revising and improving our paper. We have studied comments carefully and modified the manuscript accordingly. Detailed corrections are listed below point by point:

  1. For the miscellaneous writing faults

Concerning the miscellaneous writing faults, I have checked and corrected the faults one by one, thanks a lot for your careful review.

  1. For misused image in Fig 12

The images of Fig 12 is carefully checked, the formerly wrong image in Fig 12 is replaced with the correct one, thank you for pointing out my fault!

  1. Why are there [C,J,D,M] in brackets? Not a standard format

The [C], [J], [D], [M] symbols are from templates, [C] means conference paper, [J] means journal, [D] means dissertation, [M] represents manual.

Reviewer 3 Report

Comments and Suggestions for Authors

The draft is interesting and has potential in the area of optical manufacturing.

However, I would like to see better-resolution surface roughness images and the spectral response of elemental analysis, not low-pixel images. At the moment, readers should believe that the presented image represents a response to a particular element.

Additionally, the high noise in Figures 9 b and c makes distinguishing between signal and noise difficult. It's crucial to highlight and better present the exact link between the proposed structure and fluorescence measurement, as it's currently not clear and convincing.

The discussion part should be strengthened.

.Finally, I recommend a major revision of this draft.

Author Response

Dear reviewer, thank you for your work for our manuscript. Those comments are all valuable and very helpful for revising and improving our paper. We have studied comments carefully and modified the manuscript accordingly. Detailed corrections are listed below point by point:

  1. For low quality images issue of Fig.9 and Fig.10

Due to auto compression by text editing software(WPS), the figures are of poor quality. I re-upload the images.

  1. For discussion strengthening issue.

The discussion from line 239 to line 250 is re-organized, the spectrum results of ODC and NBOHC are discussed respectively, for noise in fluorescence spectrum, it comes from the chemical defects induced by residual part after deprivation of oxygen atoms and silicon atoms in chemical reaction, the reason is also stated from line 241 to 245.

Thank you for your valuable advices on improving my draft!

Round 2

Reviewer 3 Report

Comments and Suggestions for Authors

Authors improved the quality of their draft and answers to all referee questions. Based on that fact, I recommend accepting the revised draft in its unchanged form.